

# Antibiotic use in infants within the first year of life is associated with the appearance of antibiotic-resistant genes in their feces

Tolulope Elizabeth Fadeyi[1,*], Omolanke Temitope Oyedemi[2,*], Olushina Olawale Awe[3] and Funmilola Ayeni[4]

[1] Department of Microbiology, Faculty of Pharmacy, University of Ibadan, Ibadan, Oyo, Nigeria
[2] Department of Microbiology, Faculty of Science, Adeleke University, Ede, Osun State, Nigeria
[3] Department of Statistics, Institute of Mathematics, Statistics and Scientific Computing (IMECC), University of Campinas, Campinas, Sao Paulo, Brazil
[4] Department of Biology, Simmons University, Boston, MA, USA
* These authors contributed equally to this work.

Corresponding author
Funmilola Ayeni,
ayeni@simmons.edu

## ABSTRACT

**Background:** Antibiotic resistance, an increasing challenge, is not only a national threat but also a global threat. Carriage of resistance genes is not limited to adults alone, various microbiota niches present in the body system of children have been found to harbor bacteria carrying resistant genes, especially, their gut microbiota. This study aims to identify selected antibiotic-resistant genes from the fecal samples of infants and the association of antibiotics use with the occurrence of resistant genes in the infant's gut.

**Methods:** A total number of 172 metagenomic DNA samples previously extracted from stool samples of 28 Nigerian babies longitudinally within their first year of life were screened for the presence of ESBL genes (*bla*SHV, *bla*TEM, and *bla*CTX-M), PMQR genes (*qnrA*, *qnrB*, *qnrS*, *qepA*), ribosomal protection protein tetracycline resistance gene, (RPP) β-lactamase (*blaZ*), macrolide (*ermA*, *ermB*, *mef*A/E), aminoglycoside modifying enzymes gent$^R$ (aac(6′)/aph(2″)) and *dfrA* genes by PCR. Nineteen (19) of the 28 babies used antibiotics during the study. The association between antibiotic use by the babies within the first year of life and occurrence of resistant genes were analyzed by Spearman rank correlation.

**Results:** One hundred and twenty-two (122) samples (71%) out of the 172 isolates had antibiotic-resistance genes. PMQR genes were absent in all the samples. Three isolates had *bla*TEM gene, nine isolates had *bla*SHV gene, six isolates had *bla*CTX-M gene and 19 isolates had *dfrA* gene, 31 samples had *tet* gene, 29 samples had *mef* gene, 27 samples had *ermB* gene, four samples had *ermA* gene, 13 samples had *blaZ* gene and 16 samples had *aac* gene. The babies whose samples had resistant genes used antibiotics in the same months the samples were collected. Interestingly, the 11 babies whose samples had the *dfrA* gene all used antibiotics in the same months their samples were collected but none of them used trimethoprim/sulfamethoxazole antibiotic. The overall correlation matrix of the babies showed a strong association between antibiotic use (AU) and antibiotic use presence of resistance genes (AUPRG) with a coefficient of 0.89. Antibiotic-resistant genes are present in the gut of infants and their occurrence is strongly connected with antibiotic use by infants.

# INTRODUCTION

Approximately 700,000 deaths worldwide annually are caused by antimicrobial resistance (AMR) microorganisms and every country is potentially affected by it. By 2050, the amount could rise to 10 million *per annum* if adequate measures are not taken (*Jasovský et al., 2016*). In particular, the increased use and administration of antibiotics both for humans and livestock has produced high selective pressures in bacteria in order to have genetic elements that are responsible for their resistance to antibiotic (*Fadare et al., 2015*; *Akinbami, Olofinsae & Ayeni, 2018*; *Ayandiran et al., 2018*).

There can be a persistence of antibiotic resistance for several years even when there is no more exposure to the selective pressure caused by the use of antibiotics (*Huang et al., 2011*). In addition, the environment does not have as much antimicrobial resistance genes (ARGs) as the human gut microbiome (*Black et al., 2013*, *Fadare et al., 2015*). Also, commensal bacteria play a key role in the spreading and development of ARGs even when they are not targeted when treating infections caused by pathogenic bacteria (*Akhtar, 2009*; *Ayeni et al., 2015*). Antibiotic resistant genes present in the gut of mothers can be transferred to babies during or immediately after birth (*Aronson, 2014*; *Li et al., 2020*). The fact that these ARGs can be transferred to pathogenic bacteria from non-pathogenic bacteria present is a pointer towards a risk to human health as the human intestinal tract remains a reservoir for antibiotic resistance genes (*Bin Abdulhak et al., 2011*; *Zhang et al., 2022*). Besides, ARGs can stay in the human gut for a very long time even without their coming in contact or exposure to long-term antibiotics. A report on prevalence of resistance genes in infants during the first months following birth reports that the most frequently occurring ARG are the ESBL genes (*Herindrainy et al., 2018*). There has been a cohort study based on multisite birth in eight (8) sites in different countries of Asia, sub-Saharan Africa and South America (*Platts-Mills et al., 2015*). Also, there have been reports from cross-sectional community-based surveys and health-care settings (*Bajis et al., 2014*) on the use of antibiotics. However, there is a dearth of information on the resistant pattern in infants in a longitudinal study. It is critical to better define the prevalence and molecular composition of resistant organisms for neonates, infants, and children to adopt best-practice infection control measures and aid in the appropriate choice of empirical antimicrobial coverage for infections in these pediatric populations (*Nahla et al., 2018*).

The changes in the gut microbiota of Nigerian infants within the first year of life have been reported (*Oyedemi et al., 2022*), this study further reports the correlation between antibiotic use and occurrence of selected resistant genes in infants feces.

## MATERIALS AND METHODS

### Study population and preliminary description

The metagenomic fecal DNA used in this study was obtained from Oyedemi et al. (2022) study. In summary, the previous study was a longitudinal one of convenience sampled participants. The enrolment began at the postnatal ward immediately after birth. Fecal samples were collected monthly from recruited babies within 12 months (median age: 7 months) at the Department of Obstetrics and Gynecology, Federal Teaching Hospital, Ido-Ekiti from October, 2016 to September 2017. Ethical approval was obtained from Federal Teaching Hospital, Ido-Ekiti, Ekiti State, Southwest Nigeria, with protocol number: ERC/2016/09/29/44B. Mothers of the babies provided written informed consents for their infants to participate in the study. A total of 30 babies were initially enrolled in this study, two were lost due to attrition. The enrolment was left with 28 babies (nine males and 19 females), seven of these either provided a single sample or participated for few months, and 23 babies completed study for >7 months. The use of antibiotic and herbal products was noted among the babies. One hundred and seventy-two (172) fecal samples were collected aseptically from the diapers of the babies and metagenomics DNA was extracted from stored fecal samples according to standard procedures (Oyedemi et al., 2022).

### Amplification of *bla*SHV and *bla*TEM genes

Extracted DNA was used as template in a 25 μl multiplex reaction comprising of 12.5 μl Master mix (Inqaba, South Africa), 0.25 μl of 10 pmol/mL forward and reverse *bla*TEM primers (Table 1), 0.125 μl of 10pmol/mL forward and reverse *bla*SHV primers, 1 μl of the DNA template and 10.75 μl, of molecular graded water. The PCR reaction was performed in the following conditions: initial denaturation step at 94 °C for 5 min, denaturation at 94 °C for 30 s, primer annealing at 50 °C for 30 s, extension at 72 °C for 90 s (30 cycles) and terminal extension at 72 °C for 10 min. The amplicons were observed on a 2% gel electrophoresis where the presence of bands at 768 and 857 bp for *bla*SHV and *bla*TEM respectively confirmed amplification. Positive control obtained from the Molecular Microbiology Laboratory, Faculty of Pharmacy, University of Ibadan and negative controls were used.

### Amplification of *bla*CTX-M genes

Extracted DNA was used as template in a 25 μl reaction comprising of 12.5 μl Master mix (Inqaba, South Africa), 0.25 μl of 10 pmol/mL *bla*CTX-M forward and reverse primers (Table 1), 1 μl of the DNA template and 11 μl, of molecular graded water. The PCR reaction was performed in the following conditions: initial denaturation step at 94 °C for 5 min, denaturation at 94 °C for 30 s, primer annealing at 56 °C for 1 min, extension at 72 °C for 60 s (30 cycles) and terminal extension at 72 °C for 10 min. The amplicons were observed on a 2% gel electrophoresis where the presence of bands at 543 bp confirmed amplification. Positive control obtained from the Molecular Microbiology Laboratory, Faculty of Pharmacy, University of Ibadan and negative controls were used.

**Table 1 List of primers used.**

| Gene | Primer sequence | Amplicon size | Anneling temperature | References |
|---|---|---|---|---|
| blaTEM | F 5′-GAGTATTCAACATTTTCGT-3′<br>R 5′-ACCAATGCTTAATCAGTGA-3′ | 857 bp | 50 °C for 30 s | *Maynard et al. (2004)* |
| blaSHV | F 5′-TCGCCTGTGTATTATCTCCC-3′<br>R 5′-CGCAGATAAATCACCACAATG-3′ | 768 bp | 50 °C for 30 s | *Maynard et al. (2004)* |
| blaCTX-M | F 5′-TTTGCGATGTGCAGTACCAGTAA-3′<br>R 5′-CGATATCGTTGGTGGTGCCATA-3′ | 543 bp | 56 °C for 1 min | *Nuno et al. (2007)* |
| Dfr | F 5′-CTTGTTAACCCTTTTGCCAGA-3′<br>R 5′-TTGTGAAACTATCACTAATGGTAG-3′ | 489 bp | 55 °C for 60 s | *Rhee, Choi & Ko (2016)* |
| qnrA | F 5′-ATTTCTCACGCCAGGATTTG-3′<br>R 5′-GATCGGCAAAGGTTAGGTCA-3′ | 516 bp | 53 °C for 30 s | *Wang (2009)* |
| qnrB | F 5′-GATCGTGAAAGCCAGAAAGG-3′<br>R 5′-ACGATGCCTGGTAGTTGTCC-3′ | 469 bp | 53 °C for 45 s | *Wang (2009)* |
| qnrS | F 5′-ACGACATTCGTCAACTGCAA-3′<br>R 5′-TAAATTGGCACCCTGTAGGC-3′ | 417 bp | 53 °C for 45 s | *Wang (2009)* |
| qepA | F 5′-CTTCTCTGGATCCTGGACAT-3′<br>R 5′-TGAAGATGTAGACGCCGAAC-3′ | 720 bp | 53 °C for 30 s | *Wang (2009)* |
| aac(6′) | F 5′-GAAGTACGCAGAAGAGA-3′<br>R 5′-ACATGGCAAGCTCTAGGA-3′ | 491 bp | 59 °C for 1 min | *Choi et al. (2003)* |
| mef(A/E) | F 5′-CAATATGGGCAGGGCAAG-3′<br>R 5′-AAGCTGTTCCAATGCTACGG-3′ | 317bp | 59 °C for 1 min | *Malhotra-kumar et al. (2005)* |
| ermA | F 5′-CCCGAAAAATACGCAAAATTTCAT-3′<br>R 5′-CCCTGTTTACCCATTTATAAACG-3′ | 590 bp | 59 °C for 1 min | *Malhotra-kumar et al. (2005)* |
| ermB | F 5′-TGGTATTCCAAATGCGTAATG-3′<br>R 5′-CTGTGGTATGGCGGGTAAGT-3′ | 745 bp | 59 °C for 1 min | *Malhotra-kumar et al. (2005)* |
| blaZ | F 5′-ACTTCAACACCTGCTGCTTTC-3′<br>R 5′-TGACCACTTTTATCAGCAA-3′ | 173 bp | 59 °C for 1 min | *Martineau et al. (2000)* |
| RPP tet | Degenerate<br>*F 5′-CCIGGVCAYATGGAYTTYH TDGC-3′<br>*R 5′-CKRAARTCIGMIGGIGTRCTIA CHGG-3′ | 1.3 bp | 61 °C for 1 min | *Warburton et al. (2009)* |

**Note:**
*Degenerate oligonucleotides: D=A, G, or T; H=A, C, or T; I=A, C, G, or T; K=G or T; M=A or C; R=A or G; V=A, C, or G; Y=C or T.

## Amplification of *dfrA* genes

Extracted DNA was used as template in a 25 μl reaction comprising of 12.5 μl Master mix (Inqaba, South Africa), 0.25 μl of 10 pmol/mL *dfrA* forward and reverse primers (Table 1), 1 μl of the DNA template and 11 μl, of molecular graded water. The PCR reaction was performed in the following conditions: initial denaturation step at 95 °C for 5 min, denaturation at 95 °C for 60 s, primer annealing at 55 °C for 1 min, extension at 72 °C for 60 s (35 cycles) and terminal extension at 72 °C for 10 min. The amplicons were observed on a 2% gel electrophoresis where the presence of bands at 489 bp confirmed amplification. Positive control obtained from the Molecular Microbiology Laboratory, Faculty of Pharmacy, University of Ibadan and negative controls were used.

## Amplification of *qnrB* and *qnrS* genes

Extracted DNA was used as template in a 25 μl multiplex reaction comprising of 12.5 μl Master mix (Inqaba, South Africa), 0.25 μl of 10 pmol/mL forward and reverse *qnr*B

primers (Table 1), 0.25 µl of forward and reverse *qnr*S primers (10 pmol/mL), 1 µl of the DNA template and 10. 5 µl, of molecular graded water. The PCR reaction was performed in the following conditions, initial denaturation step at 94 °C for 5 min, denaturation at 94 °C for 45 s, primer annealing at 53 °C for 45 s, extension at 72 °C for 60 s (32 cycles) and terminal extension at 72 °C for 10 min. The amplicons were observed on a 2% gel where the presence of bands at 469 and 417 bp respectively for *qnr*B and *qnr*S respectively confirmed amplification.

Positive control obtained from the Molecular Microbiology Laboratory, Faculty of Pharmacy, University of Ibadan and negative controls were used.

## Amplification of *qnrA* and *qepA* genes

Extracted DNA was used as template in a 25 µl multiplex reaction comprising of 12.5 µl Master mix (Inqaba, South Africa), 0.25 µl of forward and reverse *qnr*A primers (Table 1), (10 pmol/mL), 0.25 µl of forward and reverse *qep*A primers (10 pmol/mL), 1 µl of the DNA template and 10.5 µl, of molecular graded water. The PCR reaction was performed in the following conditions: initial denaturation step at 94 °C for 2 min, denaturation at 94 °C for 30 s, primer annealing at 53 °C for 30 s, extension at 72 °C for 60 s (35 cycles) and terminal extension at 72 °C for 10 min. The amplicons were observed on a 2% gel where the presence of bands at 516 and 720 bp for *qnr*A and *qep*A confirmed amplification. Positive control obtained from the Molecular Microbiology Laboratory, Faculty of Pharmacy, University of Ibadan and negative controls were used.

## Amplification of *blaZ*, *ermA*, *ermB*, *mef*A/E, aac(62)/aph(222) and RPP *tet* genes

The presence of the following resistance genes were investigated in a multiplex reaction, β-lactamase (*blaZ*), macrolide (*ermA*, *ermB*, *mef*A/E), and aminoglycoside modifying enzymes gent$^R$ (aac(62)/aph(222)). Degenerated primers targeted at the ribosomal protection proteins (RPP) tetracycline resistance gene was run in a single reaction (Table 1).

For the multiplex reaction, the PCR reaction mixture contained 2.5 µL of 10× buffer, 2.5 µL of 2 mM dNTPs, 0.125 µL of Hotstar Taq, 1 µL of 25 mM MgCl$_2$, 0.25 µL of 10 pmol/mL of each primer, 15.875 µL nuclease free water and 0.5 µL of diluted DNA template making a total volume of 25 µL. The reaction condition set for the primers outlined in Table 1 was followed, initial denaturation at 95 °C for 5 min followed by 30 cycles of denaturation at 95 °C for 1 min, annealing at 59 °C for 1 min, extension at 72 °C for 1 min 30 s and terminal extension at 72 °C for 10 min. The amplicons were observed on a 1.5% gel electrophoresis. Band sizes of 173, 590, 745, 317 and 491 bp confirmed amplification for *blaZ*, *ermA*, *ermB*, *mef*A/E, and aac(62)/aph(222) genes respectively.

For the RPP single reaction, a modified method of *Malhotra-kumar et al. (2005)* was used. The PCR reaction mixture contained 2.5 µL of 10× buffer, 2.5 µL of 2 mM dNTPs, 0.125 µL of Hotstar Taq, 1 µL of 25 mM MgCl$_2$, 0.25 µL of 10 pmol/mL of the primer, 17.875 µL Nuclease free water and 0.5 µL of diluted DNA template making a total volume of 25 µL. The PCR condition was set at initial denaturation at 95 °C for 15 min, 35 cycles of

**Table 2 List of antibiotics used by individual baby at different months.**

| Babies | Antibiotics used | Age of babies when antibiotics were used (months) |
|---|---|---|
| 1 | Amoxicillin | 7, 10 |
| 2 | Erythromycin | 1 |
| 5 | Chloramphenicol | 1 |
| 6 | Amoxicillin/Clavulanic acid | 8 |
| 7 | Cefotaxime, Cefixime | At birth |
| 8 | Ampicillin/cloxacillin (Ampiclox) | At birth |
| 10 | Amoxicillin/Clavulanic acid | 3, 8, 9 |
| 11 | Amoxicillin/Clavulanic acid | 3, 9 |
| 12 | Cefuroxime, Gentamicin, Cefixime | At birth |
| 13 | Cefuroxime, Gentamicin, Cefixime | At birth |
| 14 | Amoxicillin | 7, 9 |
| 15 | Cefotaxime, Gentamicin | At birth |
|  | Amikacin, metronidazole, ciprofloxacin | 1 |
|  | Amoxicillin | 8 |
| 16 | Cefepime, gentamicin, penicillin | At birth |
|  | Ciprofloxacin | 1 |
|  | Cefuroxime (Zinnat) | 3 |
|  | Amoxicillin | 6 |
| 17 | Amoxicillin | 5 |
| 20 | Amoxicillin | 3, 6 |
|  | Chloramphenicol | 5 |
| 22 | Cefotaxime, Gentamicin | 1 |
| 23 | Cefotaxime, Gentamicin | At birth |
| 24 | Cefixime | At birth |
|  | Amoxicillin | 6 |
| 25 | Amoxicillin | 7 |
| 26 | Amoxicillin | 5 |
| 27 | Amoxicillin/Clavulanic acid | 5 |
|  |  | 7 |
| 28 | Amoxicillin/Clavulanic acid | 5 |
|  | Ampicillin/cloxacillin (Ampiclox) | 7 |

Note:
Each baby used antibiotics at varying time points from birth (indicated by 0) to 12 (indicated by 1–12) months.

denaturation at 95 °C for 1 min, annealing at 61 °C for 1 min, extension at 72 °C for 1 min 30 s and terminal extension at 72 °C for 10 min. Positive control was generously supplied by the Gut Microbiology unit of the Rowett Institute of Nutrition and Health, University of Aberdeen, Scotland, UK and nuclease free water was used as the negative control.

## Antibiotics used by the babies

The babies used antibiotics at one point or the other as they grew. A great number of the babies delivered through caesarean section all used antibiotics at birth especially the penicillin family and even though it was not documented in this study, the mothers used antibiotics immediately after delivery (Table 2).

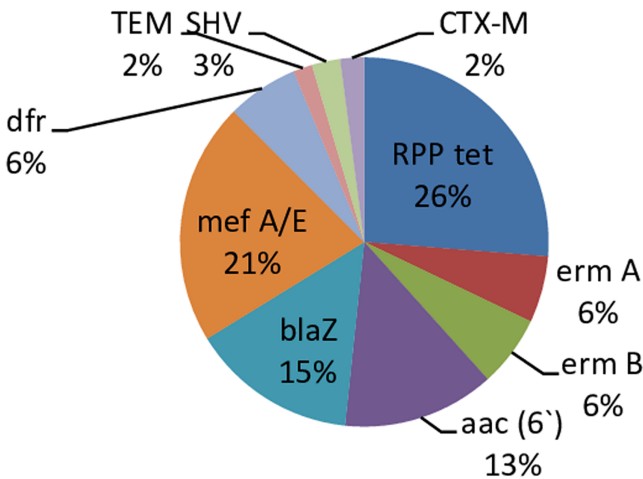

**Figure 1 Percentage occurrence of resistant genes in infants' stool.** Each cell represents the percentage occurrence of resistance genes in infants' stool over time. Tet gene has the highest, TEM and CTX-M genes had the lowest.               

## Statistical analysis

Spearman rank correlation was used to determine the level of association between antibiotic use and presence of resistance genes in the feces of the babies.

## RESULTS

A total of 172 fecal samples were obtained from 28 babies. One hundred and twenty-two (71%) out of the 172 samples had resistance genes with 50 (29%) being without the resistant genes. There was no amplification of the PMQR genes in all the isolates. The predominant resistance gene in the present study is RPP *tet* (26%) and the least occurring genes are *bla*TEM and *bla*CTX-M (2%) respectively Fig. 1.

The occurrence of specific resistance genes was prevalent in some of the babies. RPP *tet, aac, bla*Z and *mef* A/E were more abundant in baby 1, *erm*B and *bla*SHV genes in Baby 11, *dfr*A in baby 20, *erm*A in baby 23, and *bla*CTX-M in baby 24. Other babies had an even distribution of the genes (Table S1).

The Spearman rank correlation found association between antibiotic use (AU), antibiotic use presence of resistance genes (AUPRG) in 17(61%) babies, while there was no association between the two variables in 11(39%) babies. The overall correlation plot involving AU, AUPRG and NAUPRG (no antibiotic use presence of resistance genes) of the seventeen babies is shown in Fig. 2. The blue circles depict higher association between antibiotic use and presence of antibiotic genes, while the red circles depict lower correlations. The corresponding matrix of correlation coefficients of the variables is shown in Table 3. The correlation coefficient is between −1 and 1.

If the correlation coefficient value between any two variables is close to −1 or 1, then the two variables are said to be highly correlated, and it depicts that there is a high association between the two variables. In this present study, the antibiotic use and presence of resistance genes are the two variables which depict higher association.

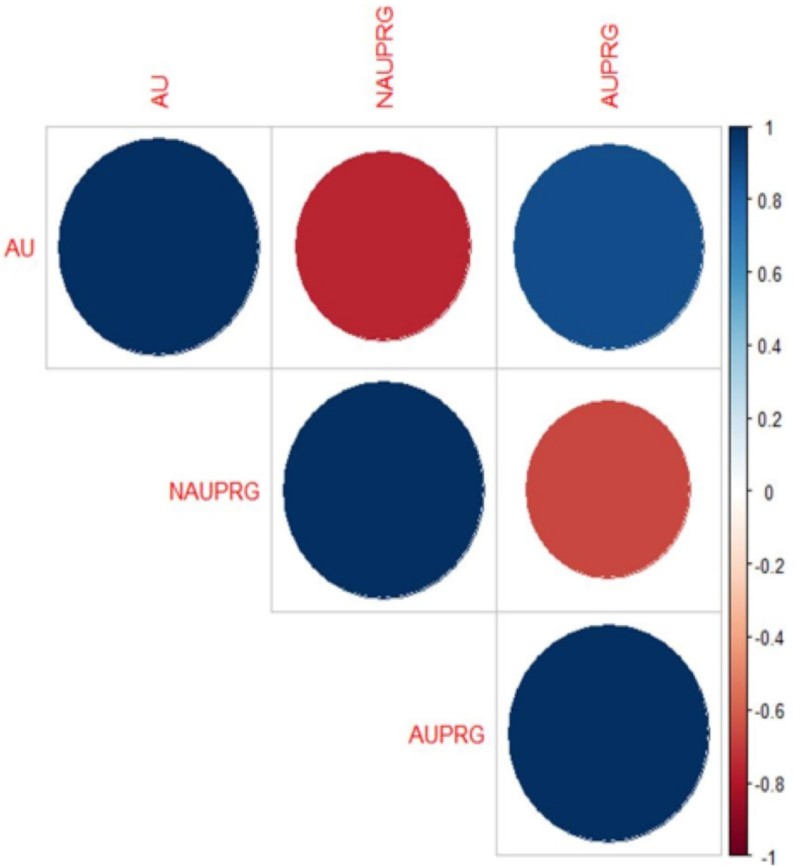

**Figure 2 Correlation plot of antibiotic use and the presence of resistance genes in infants' stool.** The big blue circle showed a positive correlation between AU and AUPRG, and the small red circle showed a negative correlation between the two parameters.

**Table 3 Correlation matrix.**

|  | AU | NAUPRG | AUPRG |
| --- | --- | --- | --- |
| AU | 1.00 | −0.76 | 0.89 |
| NAUPRG | −0.76 | 1.00 | −0.67 |
| AUPRG | 0.89 | −0.67 | 1.00 |

**Note:**
AU, Antibiotic use; NAUPRG, No antibiotic use but presence of resistant gene; AUPRG, Antibiotic use and presence of resistant gene.

Note that the specific genes driving the correlation were not investigated. Spearman's non-parametric correlation coefficient was adopted for this study because of the non-parametric binary nature of the variables (AU and AUPRG). The correlation coefficient between AU and NAUPRG is negative (−0.76). This depicts that the two variables are highly negatively associated with each other. It either means that as AU is increasing, NAUPRG is decreasing or vice-versa. AU and AUPRG maintain a higher correlation with a coefficient of 0.89 which depicts that they are highly associated. Finally,

the correlation between AUPRG and NAUPRG is also highly negatively correlated with coefficient −0.67. It shows that the two variables (AUPRG and NAUPRG) are moving in opposite directions.

## DISCUSSION

Antibiotic resistance has been a threat on a global level in all people, even in infants. Newborns have been found to harbor resistant genes and these have been linked to antibiotic use, environment, or vertical transfer from the mother to the child (*van Schaik, 2015*; *Patangia et al., 2022*). In the current study, all babies whose samples had these resistant genes used antibiotics at one time points in their first few months of life. The overuse or misuse of antibiotics can give rise to the transfer and development of antimicrobial resistance both at the population level and individual level (*Bebell & Muiru, 2014*; *WHO, 2022*).

Due to concerns regarding cartilage toxicity, fluoroquinolones are not approved in Nigeria and other parts of the world for routine use in children, although they are widely used in adults (*Choi, Kim & Kim, 2013*). None of the babies in the present study had PMQR genes. This is a pointer to the fact that non usage of the antibiotics may lead to less selective pressure and non-appearance of the genes. This result is a little bit different from the study of *Silva-Sánchez et al. (2013)* that detected some PMQR genes in isolates gotten from infants in Mexico, however, they were in a small quantity and higher in comparison to the results from China (*Han et al., 2010*), and it was similar to those from Uruguay, another Latin American country (*Garcia-Fulgueiras et al., 2011*).

Genes can be transferred horizontally from organisms of the same species or different species (*Hall et al., 2020*) thereby explaining the prevalence of *dfrA* gene in the early months of the babies even though they did not use any trimethoprim/sulfamethoxazole antibiotics (*Brolund et al., 2010*). In this study, nineteen samples (15.6%) out of 122 that had resistant genes had *dfrA* gene and is further correlated with the use of antibiotics as these 19 samples were gotten from 11 babies and all of them (except one) used antibiotics at one or more points in their first few months of life. These genes appeared immediately after use and linger on for a while before disappearing while some appeared 2 months after the use of antibiotics. The genes however all appeared after the use of antibiotics and not before the use of antibiotics. Seventeen out of the 19 isolates that had the *dfrA* gene were isolates gotten from the babies within the first 2 months of life, thereby inferring that the gene disappeared as the babies grew older even though antibiotics were still used at some points. This could be as a result of the fact that the *dfrA* gene is harbored by a group of microbes that only colonizes the gut at a very early age and gets replaced overtime. However, majority of the babies born through caesarean section used antibiotics at birth and also their mothers if the babies born through caesarean section do not use antibiotics, there is a possibility that the mother used it and these microbes harboring these resistant genes can be passed from the mother to the child through breastfeeding. This possibly means the babies got this gene from birth environment. Six out of the 11 babies that had this gene were given birth to through vaginal delivery while the remaining five is by caesarean section. During vaginal delivery, a great number of bacteria are encountered by

the infant as it passes through the birth canal thereby acquiring some microbiota as the mother (*Mitchell et al., 2020*).

Also, antibiotic resistance can be acquired through accumulation of mutations or through horizontal transfer of mobile genetic elements (*Martinez, 2014*). The prevalence of *dfrA* gene in this study can be linked to the study of *Barraud et al. (2018)* where 14 (77.8%) of the 18 integron-positive isolates gotten from stool samples of newborns were resistant to trimethoprim/sulfamethoxazole because there is presence of *dfrA* cassettes. These integrons were gotten from the mothers who apparently were administered some antibiotics during labor. This seems to be more convincing as 89.4% of *dfrA* gene seen in this study appeared from isolates gotten in the first 2 months of life. From the 37 isolates that had resistant genes, six isolates (16%) had *bla*CTX-M gene and this correlates with antibiotic use as these six isolates were gotten from five babies and all of these five babies except one used penicillin and cephalosporin families at one or more points in their first few months of life. The correlation between a variable and itself is high and it is perfect (*Awe, 2012*). One of the babies also used herbal products. Out of the six isolates that had the *bla*CTX-M gene, five were isolates obtained from the babies at different time intervals unlike the *dfrA* that was within the first 2 months of life. The months these resistant genes appeared tallies directly with the months the babies used antibiotics which satisfies the notion that there is a relationship between antibiotic use and occurrence of resistant genes. The antibiotics used by the babies include, amoxicillin/clavulanic acid, cefixime and amoxicillin. The *bla*CTX-M gene in this study has the second highest prevalence amongst other ESBL genes, however, *Hijazi & Fawzi (2016)* reported that *bla*CTX-M has the highest prevalence than other ESBL genes in infants. This is in contrast with the result of this study as *bla*SHV has the highest prevalence amongst other ESBL genes. Nine isolates (24%) out of the 37 isolates that had resistant genes had *bla*SHV gene and this is further linked to antibiotic use as these nine isolates were obtained from five babies and all of these five babies except one used antibiotics at one or more points in their first few months of life. Six out of the nine isolates that had the *bla*SHV gene were isolates obtained from the babies within the first 3 months of life. The months these resistant genes appeared tally directly with the months the babies used antibiotics which satisfies the notion that there is a relationship between antibiotic use and occurrence of resistant genes. Two variables may be highly related but not causally associated (*Awe, 2012*). Two out of the babies also used herbal products alongside the use of antibiotics but not at the same month with the use of antibiotics. The antibiotics used by the babies include, amoxicillin/clavulanic acid, cefixime, cefotaxime and ampicillin/cloxacillin and this also further confirms that the use of antibiotic has a relationship with the occurrence of resistant genes.

Among the 37 samples that had resistant genes, three (8%) had *bla*TEM gene and this is linked to antibiotic use, since only one baby produced the three samples. The months these resistant genes appeared tally directly with the months the baby used antibiotics. The antibiotics used by the babies include ampicillin/cloxacillin (ampiclox) and this also further confirms that the use of antibiotic has a relationship with the occurrence of

resistant genes. Antibiotics used by these study infants were the β-lactams, macrolides and aminoglycosides. Noteworthy in this study is the appearance of *bla*Z gene when infants are exposed to β-lactam antibiotics. From observation, this antibiotic is commonly used and prescribed and infants might have acquired the resistance genes as a result of usage. Correlations between consumption of penicillins and resistance to ampicillin have been described (*Kahlmeter, Menday & Cars, 2003*). Likewise, since this gene is plasmid-borne, there is likelihood that it may be transient. *Zain, Gaik & Rikky (2018)* reported high prevalence of *bla*Z genes in infants in Singapore. β-lactamase genes are common among Gram-negative organisms; however, some Gram-positive organisms are carriers of these genes especially *bla*Z which is carried by *staphylococci*. This genus has also been associated with *mec*A, *aac*(6′)-*aph*(2′), *erm*A, *erm*B, *erm*C, and *msr*A resistance genes (*Martineau et al., 2000*). The presence of staphylococci in the gut of these babies (*Oyedemi et al., 2022*) could be possible carriers of *bla*Z gene. In the last decades, macrolide resistance determinants such as *erm* and *mef* have been reported in *streptococci*. The genetic elements on which these genes are transferred (transposons) also carries other resistance genes conferring resistance to other antibiotics such as tetracyclines, aminoglycosides and chloramphenicol. If these taxa are present in the infant's gut, it could be responsible for the high proportion of *erm*B and *mef*A/E resistance genes observed in this study (*Varaldo, Montanari & Giovanetti, 2008*).

Tetracycline resistant gene showed the highest prevalence in this study inspite of the fact that the infants did not use tetracycline antibiotics. Tetracycline resistance genes are commonly found in livestock, environment and healthy breast-fed infants with no previous exposure to antibiotics (*Gueimonde, Salminen & Isolauri, 2006*; *Ayeni et al., 2016*). It has been suggested that early occurrence of resistant genes in the human gut is independent of infants' exposure to antibiotics but might be due to exposure to maternal and environmental microbes from delivery as found in Tet$^r$ gene which appeared early in infants and persisted within the first year of life (*Zhang et al., 2011*).

*de Vries et al. (2011)* also suggested that transfer of tetracycline resistance genes likely occurred from mother to child. Furthermore, there is a very weak correlation between herbal/natural products used and occurrence of antibiotic resistant genes, as supported by a study conducted by Gupta and colleagues that suggested bacteria are unable to easily develop resistance to the multiple and/or chemically complex phytochemicals present in plant extracts (*Gupta & Birdi, 2017*), The presence of antimicrobial resistance genes is also increasing in livestock-associated bacteria and this is driven by the widespread use of veterinary antibiotics and outdoor environment (*Woolhouse et al., 2015*; *Ayeni, Olujobi & Alabi, 2015*; *Adetoye et al., 2018*, *Ayeni, Ruppitsch & Allerberger, 2018*), Bell and colleagues also suggested that increased use of antibiotics has placed selective pressure on susceptible bacteria during the last 50 years and has favored the survival of resistant strains that are also resistant to more than one antibiotic (*Bell et al., 2014*).

## CONCLUSIONS

Use of antibiotics can directly be linked to the occurrence of antibiotic resistant genes. This study has been able to show a strong association between antibiotic use and the occurrence of antimicrobial resistant genes in the gut.

## ACKNOWLEDGEMENTS

The authors wish to acknowledge the infants who participated in this study and their mothers. We also appreciate the gut health unit of the Rowett Institute of Nutrition and Health, University of Aberdeen and the Molecular Microbiology Laboratory, Faculty of Pharmacy, University of Ibadan for the technical support.

### Funding

The authors received no funding for this work.

### Competing Interests

The authors declare that they have no competing interests.

### Author Contributions

- Tolulope Elizabeth Fadeyi performed the experiments, analyzed the data, prepared figures and/or tables, authored or reviewed drafts of the article, and approved the final draft.
- Omolanke Temitope Oyedemi performed the experiments, analyzed the data, prepared figures and/or tables, authored or reviewed drafts of the article, and approved the final draft.
- Olushina Olawale Awe analyzed the data, prepared figures and/or tables, authored or reviewed drafts of the article, and approved the final draft.
- Funmilola Ayeni conceived and designed the experiments, authored or reviewed drafts of the article, and approved the final draft.

### Human Ethics

The following information was supplied relating to ethical approvals (*i.e.*, approving body and any reference numbers):

The Federal Teaching Hospital, Ido-Ekiti, Ekiti State, Southwest Nigeria, approved the study (ERC/2016/09/29/44B).

### Data Availability

The raw data are available in the Supplemental Files.

### Supplemental Information

Supplemental information for this article can be found online at http://dx.doi.org/10.7717/peerj.15015#supplemental-information.

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
