# Peer review of "Antibiotic use in infants within the first year of life is associated with the appearance of antibiotic-resistant genes in their feces"

_PeerJ, doi:10.7717/peerj.15015_

## Round 0.1 · original submission · Major Revisions

The three reviewers gave their own suggestions, and I agree with them. Authors are requested to revise the manuscript carefully, according to the suggestions of reviewers, or give convincing reasons. Otherwise, experts have the right to reject the manuscript.

·

Basic reporting

The manuscript by Tolulope Elizabeth Fadeyi et al., reported the association between presence of antibiotics resistance genes and antibiotics usage. While the result could be interesting, there are some weaknesses that the authors need to address.

1, The most important consideration for determining the presence of antibiotics resistance genes using PCR is the false positive and the false negative.
False positives may occur when PCR may still produce amplification even using water as a template. On the contrary, the false negative result may be caused by failure of PCR amplification due to failure of DNA extraction, or other PCR procedures. How are the false positives and false negatives controlled?

2, The authors had used different primers for determining the corresponding antibiotic resistance genes. How degenerate primers have been determined?

3, Line217. “The corresponding matrix of correlation coefficients (Table 3).” Please revise the sentence.
4.Line 219. “meaning that there is high association between the variables, In the present study, the”. Please correct the sentence.

Experimental design

.

Validity of the findings

.

Reviewer 2 ·

Basic reporting

.

Experimental design

.

Validity of the findings

.

Additional comments

This paper is trying to identify the selected resistant genes from faecal samples and correlation between antibiotics use and resistant genes. From scientific novelty and dissemination point of view, it should be interesting broad readers. Some concerns need to be addressed especially my comment 4 and then get reevaluation before publishing:
1. Why you focused on these genes only, what is the logic and please detail your logics.
2. When you amplify the tet, ermA, ermB, blaZ, aac and mef genes, did you include positive and negative controls?
Line79-82, edit the language, two sentences are not allowed here.
3. Line 34 and 35, unify the 122
4. Have you checked the antibiotics treatment for the mothers before delivery or during breast feeding process? Have you checked the gut microbiomes and resistant genes of the babies’ faecal samples before the treatment of antibiotics? This is important for your conclusion.
5. The resolution of figure needs great improvements.
6. Summarize your results from line183-210 and make a figure to show your data instead of long words in the main text.

Reviewer 3 ·

Basic reporting

1. English usage should be improved by a fluent speaker.
2. references format should be consistent

Experimental design

1. The gene detected at a specific time point does not mean the baby had genes at that time. if the genes are low copy in DNA samples, it could not be detected. Authors also mentioned, mother can transfer gut microbiome at baby birth which are high possible contain the microbe with resistance genes, and resistance gene high copied and high expressed when there is selective pressure. author should re-organize language. And in experimental section, different genes are detected by different PCR methods, including template concentration, positive control template concentration, Polymerase, PCR cycles.
2. Authors can use RT-PCR to detect the copy number in DNA samples or detect the expression level in RNA samples, then the correction of antibiotics usage and resistance will be clear.
3. The method used to extract DNA is missed in experimental section.

Validity of the findings

authors should re organise language in result section and express clearly what's the conclusion.

---

## Round 0.2 · Minor Revisions

Please revise the manuscript carefully and proofread the whole text.

·

Basic reporting

I have no other comments. The authors have addressed all my concerns.

Experimental design

none.

Validity of the findings

I have no other comments. The authors have addressed all my concerns.

Reviewer 2 ·

Basic reporting

This is the 2nd time to review the manuscript and I do think the authors took serious considerations for all my previous comments. I would suggest to publish as it is.

Experimental design

This is the 2nd time to review the manuscript and I do think the authors took serious considerations for all my previous comments. I would suggest to publish as it is.

Validity of the findings

This is the 2nd time to review the manuscript and I do think the authors took serious considerations for all my previous comments. I would suggest to publish as it is.

Reviewer 3 ·

Basic reporting

Manuscript has been modified. But authors still did not release my concern about PCR experiments and results. And more minor mistakes appeared. Even the errors i pointed out last time still exist, like reference format. Authors should check throughout manuscript.
1.I am still concerned about using PCR to detect the genes. Authors mentioned their aim is to detect the gene instead of quantitive. Then could author provide agarose gel electrophoresis picture for PCR results including positive and negative control, like Fig 1 in Malhotra-kumar et al., 2005.
2. There are more mistaks and improper format in manuscript, I could not believe this is a ready publishing manuscript. eg: “95oC” in line 71-72, “10pmol/mL” in line 68 and 69. Authors should double check throughout the manuscript.
3. line 170, “The PCR condition was set at initial denaturation at 95oC for 15 mins”, it needs to be confirmed again, it is 3 mins in reference.
4. the format for same gene should consist, eg; blaCTX-M in Line 196 and blaCTX-M in line 205, author need to check carefully.
5. improper reference format and mistakes exist. eg:reference 43, 52
The format of reference is still inconsistent and mistakes . eg: reference 43.

Experimental design

no comment

Validity of the findings

no comment

---

## Round 0.3 · accepted · Accept

As a long-term clinical study, although this paper is more descriptive of the results and conclusions, the summary of clinical experience has some significance and is generally worth publishing.